# T-Measure: A Measure for Model Transferability

## Abstract

A popular paradigm in AI modeling, including computer vision, natural language processing, and graph modeling, is applying a large pre-trained model that has been fine-tuned for a particular task on novel datasets. However, many such models are published in model repositories, fine-tuned using different types of source data. Consequently, practitioners face the problem of *model selection* – choosing the best model for their task from a repository of models. Model performance in a target domain depends on factors including task definition, model architecture, data distribution, and the model transfer method. Previous model selection methods in transfer learning focus on task definition when assessing transferability, and often require a labeled dataset in the target domain. We formulate the transfer problem as label-agnostic model selection, where the goal is to choose the best-performing model on a target domain without access to labeled data. Specifically, we analyze the impact of source domain training data on model transferability. To measure this transferability, we introduce a new type of quantitative measure, the T-Measure, which correlates with the test-time performance of a model on an unlabeled target domain. We propose a T-Measure estimation method which incorporates distributional measures of the source domain's training data instances, the distribution of the target domain's instances, and the base performance of a task-specific model to create a ranking of models. We then adapt previous task-centric transferability measures for data-centric selection and compare them against T-Measure. We thoroughly evaluate the T-Measure performance for 4 tasks and 11 datasets and show its effectiveness in ranking models for model selection compared to baselines.

## 1 Introduction

The emergence of pre-trained models have led to a substantial improvement in different machine learning domains such as computer vision, natural language processing (NLP), and graph prediction. A common paradigm is that a model is first pre-trained in an unsupervised manner on a vast corpus, and afterward the resulting model parameters are used as a starting point for training (fine-tuning) models to complete various tasks using a much smaller set of labeled data. In NLP, for example, a single pre-trained model (Devlin et al., 2019) has been fine-tuned for tasks including text classification, emotion detection, question answering, inference, etc. In this paradigm, models share the same pre-trained model and differ primarily on the dataset used during fine-tuning. Practitioners seeking to reuse fine-tuned models in a new domain face the problem of *model selection* – choosing an appropriate model configuration to apply to their problem. In this paper we introduce the T-Measure, a measure of model transferability that is label-agnostic and can guide practitioners in model selection. T-Measure focuses on the model transferability from the perspective of data.

Model selection is the problem of selecting a model most appropriate for a specific task. Model selection is challenging for practitioners for several reasons. The sheer number of models available makes evaluating models a time-consuming and computationally intensive process. For example, a practitioner looking for a text classification model is faced with 780 options when searching a popular model repository, huggingface [1]. Many model selection approaches require a labeled development set to evaluate models. Researchers exploring new domains may have no labeled data

---

[1] accessed on 30 Jun 2023

available, making model evaluation difficult without a laborious labeling effort. These challenges can result in haphazard model selection conducted through trial and error. To address these model selection challenges, we propose a principled model selection approach based on unsupervised representation learning that requires no labeled data while remaining computationally efficient.

Model transferability, the performance of the model after transfer learning, is an important problem given the computational costs of training large, parameter-dense models. Recently there have been a few measures introduced to estimate the transferability of the models. A key assumption of these measures is the availability of a labeled development dataset for the target domain making them computationally expensive and incompatible in label agnostic setting (Zamir et al., 2018). Another shortcoming of these prior works is the narrowness of their evaluation, which have focused on computer vision tasks on the CIFAR (Krizhevsky et al., 2009) and ImageNet (Deng et al., 2009) datasets. The limited evaluation of these measure in the context of only few datasets raises concerns regarding the generalizability of them across different tasks and domains. In particular the effect of the source dataset on transfer is overlooked in all of those measures. Some recent body of work suggest that source dataset is an important factor in transfer: (Zhao et al., 2022) suggests that some datasets are intrinsically harder than other datasets for any task. (Ethayarajh et al., 2021) show that training datasets have different amount of useful information for trained models.

In this paper, we propose T-MEASURE as a criteria for model selection for machine learning tasks. T-MEASURE estimates the transferability of the models from the dataset perspective. It utilizes unsupervised representation learning to gain insight from datasets without requiring any label. It leverages the representation learning to quantify the effect of datasets during transfer. To the best of our knowledge T-MEASURE is the first data-centric transfer measure in zero-shot transfer. We adapt the previously used transfer measures to the data-centric zero-shot setting, compare T-MEASURE with them, and evaluate T-MEASURE on 4 different tasks and 11 different datasets. We show that T-MEASURE has better performance compared to other transfer measures and is more robust. Our contributions are:

- We introduce the novel problem of label-agnostic model selection.
- We present T-MEASURE as a transfer measure.
- We use representation learning and introduce a method to compute T-MEASURE in a zero-shot transfer setting.
- We adapt the previous transfer measures to the zero-shot constraint.
- We analyze the performance of T-MEASURE among 4 different tasks and 11 datasets.

## 2 PROBLEM DEFINITIONS

In this section, we formally define the problem of model selection in zero-shot transfer learning. First, we introduce the general problem of model selection. Second, we focus on the model selection and its challenges in transfer learning. Then we introduce transfer measure for model selection. Finally, we scope the problem by focusing on the model selection in zero-shot transfer learning.

### 2.1 MODEL SELECTION

Suppose $\Phi = \{\phi_i\}_{i=1}^{n}$ is a set of $n$ models, each trained on a task $T_i$. Each task $T_i$ is represented by a labeled dataset $D_i = D_i^{train} \bigcup D_i^{test}$ and has an evaluation metric $E_i$ to measure its performance. Let $\beta_i$ be the architecture of the model $\phi_i$. Therefore each trained model is identified with three variables $(T_i, D_i, \beta_i)$ and its performance is denoted by $E_i(T_i, D_i^{test}, \phi_i)$.

Let $T_{trg}$ be the target task with dataset $D_{trg}$ and evaluation metric $E_{trg}$. In general, model selection for $T_{trg}$ is the problem of selecting a model $\phi^* \in \Phi$ which has the best performance on target:

Table 1: Symbol descriptions

| Symbol | Description |
|--------|-------------|
| $D_x$ | Dataset x |
| $D^{train}$ | Training subset of the dataset D |
| $T_x$ | Task x |
| $E_i$ | Evaluation metric $i$ |
| $\beta_x$ | Model architecture x |
| $\alpha_i$ | Transfer method $i$ |
| $\phi$ | Model |
| $R_D$ | Representation space based on dataset D |

$$\phi^* = \underset{\phi \in \Phi}{Argmax}\, E_{trg}(T_{trg}, D_{trg}^{test}, \phi) \tag{1}$$

## 2.2 Model Selection in Transfer

In this subsection, we specify the model selection problem in a transfer setting. Let $T_{trg}$ be a target task with labeled dataset $D_{trg} = D_{trg}^{train} \bigcup D_{trg}^{test}$. Let $\alpha$ be a model transfer method characterized as a function that transfers a model $\phi$ based on the target task and dataset and creates a new model $\phi' = \alpha(T_{trg}, D_{trg}^{train}, \phi)$. Figure 1 shows parameters of the transfer method.

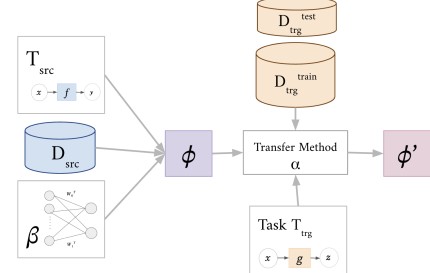

The model selection problem in this transfer setting becomes the problem of finding a model $\phi^* \in \Phi$ which shows the best performance after transfer on a target dataset. Ideally the selected model is $\phi^*$:

Figure 1: Components in model transfer from a source model to a target model

$$\phi^* = \underset{\phi \in \Phi}{Argmax}\ E_{trg}(T_{trg}, D_{trg}^{test}, \phi') \tag{2}$$

In reality, $D_{test}^{trg}$ is not accessible during model selection. However, we assume that $D_{test}^{trg}$ and $D_{train}^{trg}$ are sampled from the similar underlying distribution. Therefore, model performance on $D_{test}^{trg}$ is correlated with $D_{train}^{trg}$. And model selection chooses the model $\hat{\phi}$ with best performance on $D_{trg}^{train}$, i.e:

$$\hat{\phi} = \underset{\phi \in \Phi}{Argmax}\ E_{trg}(T_{trg}, D_{trg}^{train}, \phi') \tag{3}$$

Where $\phi'$ is the model $\phi$ transferred with method $\alpha$ using $D_{trg}^{train}$ dataset for the task $T_{trg}$.

## 2.3 Transfer Measure

In this subsection, we define an abstract transfer measure for model selection in a transfer setting, and describe our approach in 3. A transfer measure is a proxy that estimates the relative performance of different models on a target dataset after model transfer. The value of a transfer measure assigned to models, facilitates the selection of the best performing model on target dataset. Without loss of generality, we identify a transfer measure as a function of a model $\phi$, a target task $T_{trg}$ and dataset $D_{trg}$. Ideally, the value of a transfer measure is highly correlated with the performance of the transferred model $\hat{\phi}$ on the target.

$$Transfer - Measure(T_{trg}, D_{trg}, \phi) \propto E_{trg}(T_{trg}, D_{trg}, \phi') \tag{4}$$

where $E_{trg}(T_{trg}, D_{trg}, \phi') \in \mathcal{R}$ is the value indicating the performance of $\phi'$ on the target. Ideally, a transfer measure enables us to compare the performance of different models in a transfer. For example, for the two transferred models $\phi_1, \phi_2$ the inequality:

$$Transfer - Measure(T_{trg}, D_{trg}, \phi_1) \leq Transfer - Measure(T_{trg}, D_{trg}, \phi_2) \tag{5}$$

means that $\phi_2$ is predicted to have a better performance on target task $T_{trg}$ and dataset $D_{trg}$ and is a better choice for model selection compared to $\phi_2$.

In general, a transfer measure has four confounders: dataset, task, transfer method and model architecture. Figure 2 shows examples for confounders. In this paper, we focus on the dataset confounder and assume the other confounders are invariant, i.e. we isolate the dataset confounder to assess dataset effect on transfer. This assumption causes the proposed transfer measure to rely on the training dataset characteristics which are easier to quantify.

## 2.4 Model Selection in Zero-shot Transfer

We specify the model selection problem in zero-shot transfer when the only variable confounder of transfer is dataset. Let $T_{trg}, D_{trg}, E_{trg}$ be a target task, dataset and evaluation metric respectively. Let $\Phi = \{\phi\}_i$ be a family of models with the same architecture $\beta$ and task $T_{src}$ where each is trained on $D_i$. Let $\alpha$ be a transfer method. We represent $\phi'_i = \alpha(T, D_{trg}^{train}, \phi_i)$ as the transferred model by the method $\alpha$ using $D_{trg}^{train}$. In this paper, the transfer method is zero-shot: $D_{trg}^{train}$ is unlabeled and $\alpha$ is the identity function, i.e., $\phi'_i = \alpha(T, D_{trg}^{train}, \phi_i) = \phi_i$. Furthermore, models are trained,

transferred and evaluated for the same task i.e,: $T_{src} = T_{trg}$, $E_{src} = E_{trg}$. For simplicity we denote task and evaluation metric with $T$, $E$ respectively. Ideally the model selection chooses $\phi^* \in \{\phi\}_i$ : $\phi^* = \underset{\phi \in \Phi}{Argmax} \, E(T, D_{trg}^{test}, \phi)$. In reality, $D_{trg}^{test}$ is not available in transfer and assume that $D_{trg}^{test}$ and $D_{trg}^{test}$ are from the same distribution. Consequently $E(T, D_{trg}^{test}, \phi) \propto E(T, D_{trg}^{train}, \phi)$ and the model selection, finds $\hat{\phi}$:

$$\hat{\phi} = \underset{\phi \in \Phi}{Argmax} \, E(T, D_{trg}^{train}, \phi) = \underset{\phi \in \Phi}{Argmax} \, Transfer - Measure(T, D_{trg}^{train}, \phi) \qquad (6)$$

## 3 MODEL

In this section, we introduce T-Measure, a transfer measure for zero-shot model transfer. T-Measure is data-centric, i.e. it is characterized by a pair of datasets: a source dataset $D_{src}$ which the model is trained on and a target dataset $D_{trg}$. Intuitively T-Measure assesses the similarity of datasets $D_{src}$ and $D_{trg}$ in terms of a task. We propose a representation

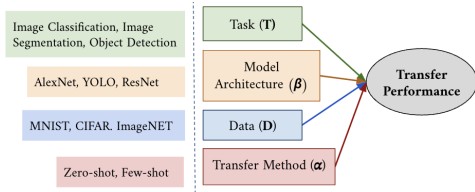

Figure 2: Confounders of transferred model performance with examples on the left side.

space that quantifies characteristics of datasets and hypothesize that it determines datasets transferability. We introduce a self-supervised representation learning approach to achieve a representation space for $D_{src}$. Then, we utilize it to compute T-Measure for task-agnostic model selection in a zero-shot setting. More specifically the task, model architecture and transfer method are invariant in this transfer setting. By using the learned representation space, we identify a subset of datapoints from $D_{src}$ which have similar characteristics to $D_{trg}$. We estimate T-Measure using the identified subset. The components of our method, creating source dataset representation space and T-Measure estimation, are shown in Figures 3, 4 respectively.

### 3.1 SOURCE DATASET REPRESENTATION SPACE

In this section, we introduce a representation space for decomposable datasets. A dataset $D$ is decomposable if we can represent an instnace $P$ in the dataset as a union of smaller units (datapoints). Moreover, datapoints in each instance are dependent on each other and independent of datapoints in other instances.

For each source dataset, we identify a representation space at datapoint level. The identified representation should have the following objectives to capture dataset structure and specifics:The relative distance of datapoints should be preserved in this representation space. The representation should be smooth, i.e. a small difference in datapoints should result in a small difference in their representation. Dependency of datapoints should be manifested as distance in their representations, i.e. dependent datapoints (datapoints in the same instance) should be closer to each other compared to independent datapoints. We say a representation is aligned with a dataset if it meets these conditions.

**Dataset Representation Alignment** We describe our approach for finding a representation space which characterizes a decomposable dataset $D$. Our approach leverages contrastive learning (Liu et al., 2021) to model the dependencies in the dataset. Specifically, we construct triplets to capture the relation of datapoints in the dataset. Then we use the triplet loss objective(Schroff et al., 2015) to find a representation space aligned with these triplets. In other words, given an initial representation space $R$ and a dataset $D$, an aligned representation space is created by fine-tuning the space with the triplet loss objective in the dataset. Let $D = \bigcup P_i$. We describe the triplet construction process for $D$ in the next paragraph.

**Triplet Construction** A triplet $(a, p, n)$ contains three points from $D$: anchor, positive, negative. Datapoints in a triplet are sampled with the goal of making the distance of the anchor and the positive sample relatively less than distance of the anchor and the negative sample. The goal of the triplet construction step is to sample triplets that capture the dependency structure of datapoints. Therefore given a pair of dependent datapoints $(a, p')$, we create a triplet by adding an independent datapoint to the pair. Since $D$ is decomposable, triplets are: $\{(a, p, n) | a, p \in P_i, n \in P_j\}$.

Ideally in an aligned space with $D$, the relative distance constraint of triplets is satisfied, i.e. for any triplet $(a, p, n) : d(a, p) < d(a, n)$. Triplet loss objective is a proxy to estimate this distance

constraint among a set of triplets in a representation space $R$. For a triplet $(a, p, n)$:

$$TripletLoss = Max(||R(a) - R(p)|| - ||R(a) - R(n)|| + \epsilon, 0) \quad (7)$$

where $R(a)$ is the representation of the anchor, $||.||$ is a distance metric and $\epsilon$ is a margin. For a dataset $D$ and a representation space $R$, we identify an aligned representation space($R_D$) by minimizing the triplet loss objective on triplets from $D$. Figure 3 shows steps of this process.

### 3.2 T-MEASURE ESTIMATION

In this subsection, we describe T-Measure estimation in zero-shot transfer for decomposable datasets. Our method leverages dataset aligned representation spaces from the previous subsection to represent $D_{src}$ and $D_{trg}$. We identify a subset of $D_{src}$ which is highly similar to $D_{trg}$. Then, we compute T-Measure by assessing the effect of this subset in the trained model. For every model $\phi_i$ (identified by $(T, D_{src}, \beta)$) and target dataset $D_{trg}$ and the representation space aligned with the source dataset $R_{src}$, T-Measure is estimated via the following steps:

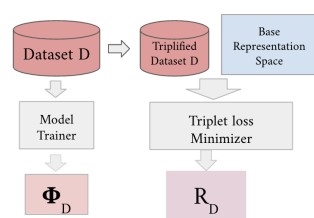

Figure 3: The steps for creating an aligned representation space for a data source D. $\phi_D$ is the model trained on $D$ and $R_D$ is the representation space aligned with $D$.

**Step 1:** Intuitively the performance of a trained model is close for similar input datasets. Also, similar subsets of training data are expected to have similar effects on a model during training. Therefore the task of estimating the effect of $D_{trg}$ on a trained model is reduced to finding a subset of $D_{src}$ similar to $D_{trg}$ and computing its effect on the trained model. In this step, we seek a subset $S_{src,trg}$ from the source dataset which is similar to the $D_{trg}$, i.e.:

$$S_{src,trg} = \bigcup_{d \in D_{trg}} \{s^* = \underset{s \in D_{src}}{Argmin} ||R(d) - R(s)||\} \quad (8)$$

Where $R$ is the representation space used for datapoints. We use the $R_{src}$ which is the $D_{src}$ aligned representation space in finding the similarity for T-Measure estimation since this representation space captures the structure of the $D_{src}$ and is more accurate in finding similar datapoints.

**Step 2:** We use $S_{src,trg}$, the most similar subset of the source dataset to the target dataset, to estimate T-Measure. We compute T-Measure by estimating the effect of $S_{src,trg}$ on the model during training. In (Ethayarajh et al., 2021), the authors introduced *V-Usabiltiy* and *Pointwise V-Information(PVI)*. We briefly describe PVI in the next paragraph. First, we describe predictive entropy then we describe PVI.

**Definition** Let $X, Y$ denote random variables with sample spaces $\mathcal{X}, \mathcal{Y}$ respectively. Let $\theta$ denote a null input that provides no information about $Y$. Given a predictive family $V \subset \Omega = \{f : \mathcal{X} \bigcup \theta \to P(Y)\}$, the predictive V-entropy($H_V(Y)$) and the conditional V-entropy are:

$$H_V(Y) = inf_{f \in V} E[\log_2 f[\theta](Y)] \quad (9)$$

$$H_V(Y|X) = inf_{f \in V} E[\log_2 f[X](Y)] \quad (10)$$

where $f[X]$ and $f[\theta]$ produce a probability distribution over the labels.

**Definition** Given random variables $X, Y$ and a predictive family V, the pointwise V-information (PVI) of an instance $(x, y)$ is:

$$PVI(x \to y) = -\log_2 g[\theta](y) + \log_2 g'[x](y) \quad (11)$$

where $g \in V$ s.t. $E[\log g[\theta](Y)] = H_V(Y)$ and $g' \in V$ s.t. $E[\log g'[X](Y)] = H_V(Y|X)$.

They used V-Usabiltiy and Pointwise V-Information(PVI) as a proxy to estimate the difficulty of datapoints and their effect on a model during training. V-usability of a random variable is defined as the expected change in the predictive entropy of the label distribution caused by conditioning on that random variable, i.e. how much information the random variable contains and how it changes the label distribution. Pointwise V-Information computes the change of label distribution predictive entropy given a single datapoint. Therefore, more difficult datapoints have less V-Usable information and have less effect on the trained model. Intuitively higher Pointwise V-Information (PVI) of

$S_{src,trg}$ on the trained model indicates the higher similarity of $\phi$ (trained on $D_{src}$) to a hypothetical model trained on $D_{trg}$. The T-Measure value of the $D_{trg}$ is achieved by finding the average PVI of $S_{src,trg}$ for model $\phi$. Figure 4 shows this process.

$$\text{T-Measure}(T, D_{trg}, \phi) = \frac{\Sigma_{(x,y) \in S_{src,trg}} PVI_\phi(x \to y)}{|S_{src,trg}|} \tag{12}$$

## 4  EVALUATION

In this section we describe our experiments to evaluate the performance of T-Measure. First, we describe implementation details common to all experiments. Then, we describe evaluation metrics. Finally, we describe the model selection experiment scenario to evaluate T-Measure.

Figure 4: T-Measure estimation step 1: given $D_{trg}$ and a model trained on $D_{src}$. $S_{src,trg}$ is the subset of $D_{src}$ which is similar to $D_{trg}$.

### 4.1  IMPLEMENTATION

As described in section 3.1, T-Measure requires a dataset specific representation space for each source dataset. We describe the process and parameters used to identify dataset specific spaces for source datasets introduced in Table **??**. The introduced source datasets are all decomposable to documents. For example, DailyDialog is perfectly decomposed to independent conversations. Furthermore, the decomposed documents have a sequential structure and can be viewed as a sequence of smaller dependent units, i.e. sentences in paragraphs or utterances in the conversations. We represent each document $d \in D_{src}$ as a sequence. For example, a conversation can be represented as $u_1, u_2, ..., u_n$ where $u_i$ is the $i^{th}$ utterance. Let $D_{src}$ be an arbitrary source dataset, and $d = u_1, ..., u_n$ be a document in $D_{src}$. Following (Zhou et al., 2022b), for every pair $(u_i, u_{i+1})$, we create 20 triplets $(u_i, u_{i+1}, u')$ by randomly sampling $u'$ from other documents in that dataset i.e $u' \in d' \in D_{src} - \{d\}$. After creating triplets for the $D_{src}$, we use SentenceBERT(Reimers & Gurevych, 2019) for the initial representation of the utterances (or sentences in non-conversational datasets). We use the Sentence-Transformer library [2] and triplet loss objective to fine-tune the sentence-transformer. We continue the process for 5 epochs. We repeat the same process for source datasets in Table **??** and identify an aligned representation space for each of them.

### 4.2  EVALUATION METRIC

The principal goal of T-Measure is to provide a quantitative score for trained models that helps with selecting the best model for a target dataset. In other words, the relative value of T-Measure for trained models should correlate with their performance in the target dataset. Therefore, we evaluate T-Measure as a model selection and ranking criteria. We evaluate the ranking of models based on T-Measure and compare it to the ranking of the model performances on the target

Table 2: Dataset Statistics. The last column shows tasks for which the dataset is labeled.

| Dataset | Dataset Size | Avg Doc Size | Tasks |
|---|---|---|---|
| PersonaChat | 19,893 | 14.8 | RS |
| Casino | 1030 | 11.6 | RS |
| MuTual | 6,371 | 4.7 | RS |
| DailyDialog | 11,118 | 8 | RS, ERC |
| Empathetic | 24,850 | 4.3 | RS, ERC |
| Friends | 897 | 14 | RS, ERC, QA |
| CIDER | 807 | 15 | QA |
| DREAM | 6,444 | 6.3 | QA |
| DialogRE | 1,788 | 12.9 | RC |
| ReDocRED | 3,053 | 7.9 | RC |
| DDRel | 6,300 | 8.4 | RC |

set. We use the Kendall$-\tau$ correlation to evaluate the ranking, computed as:

$$Kendall - \tau = \frac{\#Concordant\ Pairs - \#Discordant\ Pairs}{\#\ Pairs} \tag{13}$$

Therefore higher value of Kendall$-\tau$ is an indicator of better ranking. We also compare the $F1$ of the best selected model by each transfer measure and compare it against the ground truth in Figure 6.

### 4.3  MODEL SELECTION EXPERIMENT

We evaluate the performance of T-Measure as a transfer measure. We compare the performance of T-Measure with two baselines and a set of transfer measures created based on previous task-centric

---

[2] www.sbert.net

Figure 5: Kendall-$\tau$ of model ranking based on transfer measures. From left to right transfer measures in each plot are: Naive, V-Usability, PARC, T-MEASURE.

transfer measures A.2, i.e. PARC based models Bolya et al. (2021). Naive baseline for model selection, chooses the model with the best performance on the corresponding source dataset. The V-UsabilityEthayarajh et al. (2021) baseline chooses the source dataset with highest V-Usability value which is interpreted as the easiest source dataset for the given task. We conduct the experiment for 4 tasks:

**Emotion Recognition in Conversation (ERC)** is the task of assigning an emotion label to each utterance in a conversation. Since the datasets for ERC were annotated using different granularity of emotions, we relabeled the emotions to basic emotions (Ekman et al., 1999) using the feeling wheel (Willcox, 1982). Therefore the task is modeled as assigning an emotion label from {"happy", "sad", "anger", "surprise", "fear", "disgust", "no emotion"} to each utterance.

**Relation Classification (RC)** is the task of assigning a relation type between two named entities given a document. We selected the subset of relationships common in available datasets. These relationships are {"spouse", "sibling", "boss", "child—parent", "girl/boyfriend", "other"}. We merged the rest of the relations in each dataset to the "other" relation class.

**Question Answering (QA)** includes a document and a set of questions which can be answered based on the given document. In this paper, we modeled this task as the task of choosing the correct answer from a set of provided options.

**Response Selection (RS)** is the task of choosing the next utterance in a conversation based on the conversation history. We created response selection datasets for experiments from available conversation datasets: For each utterance in a dataset we randomly sampled 5 utterances from other conversations and used them as wrong options while the next utterance was the correct option. To ensure consistency, we adopt a uniform architecture for all task models, treating each task as a classification problem. In Question Answering and Response Selection, we format the training data as a binary classification task, i.e. given a context, a question (or current utterance) and an option, the task is to determine whether the given option is correct or not. For each task, we trained models with similar architecture and training parameters on different source datasets. We used the publicly available BERT based architecture for all models. The details of the performance of these models on different datasets are available in the appendix A.4. To evaluate the model selection based on different transfer measures, we include at least three target datasets for each task. Table **??** contains statistics of these datasets. More details are available in the appendix A.1. In this experiment, the probe set of target datasets consists of 100 instances. The results of this experiment are presented in Table 3, where we report the Kendall-$\tau$ of model rankings based on different measures. We also present the relative performance of the selected model compared to the ground truth in Figure 6.

## 5 ANALYSIS

Table 3 shows the results of the model selection experiment. The key observations are:

**Average Kendall-$\tau$ of T-MEASURE is always positive**. We observe that T-MEASURE average Kendall$-\tau$ is always positive for all tasks, i.e. the ranking based on T-MEASURE robustly contains more concordant pairs compared to discordant pairs.

**T-MEASURE ranks better than Naive method.** Naive method, ranks models based on their performance on the source dataset. This observation suggests the benefit of using a T-Measure instead of the Naive method.

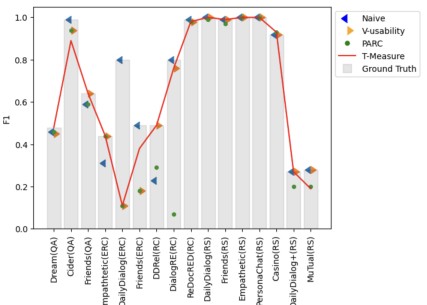

Figure 6: F1 score of the best selected model based on transfer measures.

**T-MEASURE achieves the best transferability estimation and ranking model for three tasks.** We observe that for three tasks of Response Selection, Emotion Recognition and Question Answering T-MEASURE achieves the best transferability estimation and ranking of models with Kendall-$\tau$ of 0.14, 0.33, 0.77.

**T-MEASURE average performance is at least as good as V-Usability** We observe that T-MEASURE

Table 3: The result of model ranking based on different transfer measures. The reported numbers are the Kendall-$\tau$ of the ranked models based on the given transfer measure compared to the ranking of models based on their performance on the target dataset. The size of the probe set in this experiment for targets is 100. The PARC model uses SentenceBERT for data representation.

| Task | Target Dataset | Naive | V-Usability | T-Measure | PARC |
|------|----------------|-------|-------------|-----------|------|
| **Response Selection** 6 Models | DailyDialog | **0.4** | -0.8 | 0.0 | -0.2 |
| | Friends | -0.2 | **0.6** | 0.0 | 0.0 |
| | Empathetic | 0.0 | -0.4 | **0.8** | -0.6 |
| | PersonaChat | 0.2 | -0.6 | 0.0 | **0.6** |
| | Casino | **0.6** | -0.6 | 0.2 | 0.4 |
| | DailyDialog++ | 0.0 | 0.4 | **0.6** | -0.2 |
| | MuTual | 0.0 | **0.4** | -0.6 | -0.2 |
| | AVG $\tau$ | **0.14** | -0.14 | **0.14** | -0.02 |
| **Emotion Recognition** 3 Models | Empathetic | **0.33** | -0.33 | -0.33 | -1.0 |
| | DailyDialog | -0.33 | 0.33 | 0.33 | **1.0** |
| | Friends | 0.33 | -0.33 | **1.0** | -1.0 |
| | AVG $\tau$ | 0.11 | -0.11 | **0.33** | -0.33 |
| **Question Answering** 3 Models | Dream | 0.33 | **1.0** | **1.0** | 0.33 |
| | CIDER | **1.0** | 0.33 | 0.33 | 0.33 |
| | Friends | 0.33 | **1.0** | **1.0** | 0.33 |
| | AVG $\tau$ | 0.55 | **0.77** | **0.77** | 0.33 |
| **Relation Classification** 3 Models | DDRel | 0.66 | **1.0** | **1.0** | 0.66 |
| | DialogRE | 0.0 | 0.33 | 0.33 | **1.0** |
| | ReDocRED | 0.0 | 0.33 | 0.33 | **0.66** |
| | AVG $\tau$ | 0.22 | 0.55 | 0.55 | **0.77** |

average performance is at least as good as V-Usability and Naive method in all tasks. Its performance is similar to V-Usability in Question Answering and Relation Classification. However it shows a better average performance on Emotion Recognition (+0.44 in $\tau$), and Response Selection (+0.28 in $\tau$), which supports the idea of selecting a subset of the source dataset based on the target dataset and computing their V-Usability can lead to better transferability estimation.

**T-Measure is more robust compared to Naive method.** We observe that Naive method is only creating a reasonable ranking of models for the task of Response Selection. While it fails in ordering the trained models for more challenging tasks of Emotion Recognition, Question Answering and Relation Classification. This failure is more prominent in the scenario where the available trained models have high variability in their reported source performance. For example, in the task of Emotion Recognition, the model trained on the DailyDialog dataset achieves 80% accuracy while the models trained on Empathetic and Friends datasets achieve 44% and 38% accuracy on their corresponding sources. However DailyDialog is not always the best performing model for a new target dataset. Tables 4, 6 contain the model ranking scenario for the Emotion Recognition task which provides justification for employing a T-Measure rather than ranking based on model performance on source datasets. Although the naive model performs better than random ranking for all tasks on average. Refer to the appendix A.4 for the table of model accuracy for all datasets and tasks.

**T-Measure presents less performance variation in comparison to PARC.** We observe that T-Measure presents less variation compared to PARC. Figure 6 presents boxplots of ranking performance of the methods. We observe that except Relation Classification tasks, the interquartile range of T-Measure is above the other measures. Moreover the interquartile range of T-Measure is smaller than PARC across Response Selection, Emotion Recognition and Question Answering indicating a more robust performance compared to the PARC family. We observe that PARC based transfer measures do not show consistent advantage over T-Measure in Table 12. Though they have a better performance on Relation Classification, their performance on the rest of tasks is not competitive to T-Measure. We believe this issue is mainly rooted in the zero-shot characteristics of our problem.

**Relative F1 score of the selected model is high for most measures. Relative F1 score of T-Measure based selected model is always better than PARC.** We observe that for the Response Selection task, the reported relative F1 is high for all the methods. This is mainly because of the high performance of all source models on all targets. In other words, the difference between performance of the best model and other models for every target was not high which resulted in high values of relative F1 for all methods. Therefore, for the Response Selection task, not choosing the

best performing model has negligible impact. However in Emotion Recognition, Question Answering and Relation Classification, the performance of the best performing model is higher compared to the other methods. We observe that for Question Answering and Relation Classification, model selection based on T-Measure has resulted in getting a model with relative F1 $> 0.9$. Which makes T-Measure a more reliable criteria compared to PARC based methods. Furthermore, we can conclude that for all tasks except Emotion Recognition, T-Measure failure cases in choosing the best performing model happens when the difference in performance of the best model and another model is not high. T-Measure failure in Emotion Recognition is mostly due to the difference in label distribution of the datasets, i.e. the DailyDialog dataset frequently includes "no emotion" label whereas the Empathetic dataset rarely has a "no emotion" label.

## 6    RELATED WORK

As pre-trained models and transfer learning gain traction, the issue of model selection in transfer learning has recently garnered significant attention. Transferability is a complex combination of model parameters (Jiang et al., 2019b; Yang et al., 2022), training, task definition, and dataset Sinapov et al., 2015. A number of contemporary studies have concentrated on evaluating and estimating the transferability of models across various transfer settings. Specifically, (Bolya et al., 2021; Kornblith et al., 2018) examined trained models to discern features that optimize transfer capabilities. (Bolya et al., 2021) introduced a scalable framework designed to predict the accuracy of a model on a specific dataset post-fine-tuning, while (Kornblith et al., 2018) demonstrated a correlation between the accuracy of pre-trained and fine-tuned models.

Another body of work, including (Bao et al., 2019; Nguyen et al., 2020; Tran et al., 2019), has zeroed in on quantifying and characterizing different tasks, with (Achille et al., 2019) shedding light on task transfers in particular. Among these, (Bao et al., 2019) introduced an innovative metric, the H-score, which provides a straightforward evaluation method for gauging the efficacy of transferring representations between tasks in classification contexts. LEEP, as detailed in (Nguyen et al., 2020), offers a quantitative metric for the ease of transferring knowledge between classification tasks. (Tran et al., 2019), on the other hand, evaluates the complexity of supervised classification tasks by analyzing label statistics as if they were random variables. Additionally, (Albalak et al., 2022) established a benchmark specifically for task transfers in dialogue systems. Recently, (Tan et al., 2021) introduced a transferability measure which takes into account data difficulty and task difficulty.

Previous transfer measures, are built on the assumption of having enough labeled data in the target task and domain. However, in the real world, there often isn't readily available annotated data, or creating such datasets can be prohibitively costly (Tan et al., 2018). This situation highlights the practical significance of transfer learning, particularly in cases where labeled data is scarce or expensive to obtain. A body of work Ben-David et al. (2010); Huang et al. (2021); Li et al. (2020) has focused on methods for unsupervised domain adaptation. More recent works, such as (Huh et al., 2016; Yan et al., 2020; Zhao et al., 2022; Ilyas et al., 2022), delved into quantifying the attributes of the source dataset and its influence on transferability. For instance, (Yan et al., 2020) developed a data server that identifies the most pertinent subset of source data for a novel dataset, and (Zhao et al., 2022) investigated the specific characteristics of data that enhance its suitability for few-shot learning and questioned if these are independent of adaptation methods.

Recent research has been looking at pre-trained models and the representation they learn (Peters et al., 2019). More specifically, it's trying to understand and predict how these models behave in transfer learning(You et al., 2021; Jiang et al., 2019a; Peters et al., 2019).

## 7    CONCLUSION AND FUTURE WORK

In this paper, we defined the problem of data-centric transfer estimation in zero-shot transfer. We introduced T-Measure and proposed the method to estimate it. We conducted experiments to show the effect of using the introduced transfer measures for the model selection task. In the future, we plan to expand the T-Measure estimation to estimate transferability based on other transfer confounders. In particular, we plan to introduce a general transfer measure and model selection tool which can help people with choosing the best model based on their conditions and requirements.

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

# A APPENDIX

## A.1 DATASETS

**Daily Dialog** (Li et al., 2017) is a dyadic text dataset containing conversations like typical daily communication. The conversations are short and focused on a specific topic. The conversations in this dataset have utterance level annotation for the task of emotion recognition and dialog act. The annotation for the emotion recognition are based on six basic emotions(Ekman et al., 1999).

**Daily Dialog++** (Sai et al., 2020) is an extension of Daily Dialog that contains incorrect response options for each utterance. The incorrect responses are two types i.e. random and adversarial. Random incorrect responses are utterances randomly selected from other conversations. Adversarial incorrect responses are created by human annotators.

**Friends**(Zahiri & Choi, 2017) is a dataset based on the popular sitcom with the same name. This dataset contains transcript of conversations among actors and actresses for all 10 seasons of the show. The dataset is annotated for several tasks: emotion recognition, causal entailment, question answering, etc.

**Empathetic dialogs**(Rashkin et al., 2018) includes conversations with dialog level emotion annotation. The emotion labels include 32 classes.

**Topical chat**(Gopalakrishnan et al., 2019) is a conversational dataset, where each conversation is focused on one topic selected from 8 specific topics. The dataset is created via MTurk.

**Persona chat**(Zhang et al., 2018) is a dialog dataset collected via MTurk. In each conversation, speakers are given a set of sentences called persona and are asked to talk and get to know the other speaker.

**CaSiNo**(Chawla et al., 2021) is a conversational dataset where speakers are negotiating for resources in a limited domain. Each participant is given a set of requirements and preferences and negotiates with another participant for resources such as Water. It includes utterance level annotation for the participant strategy for negotiation.

**Reflect**(Zhou et al., 2022a) is a dialog dataset created based on the one-to-many property of conversations. It includes 5 different inference types and responses based on them for the utterances collected by human.

**MuTual**(Cui et al., 2020) is a conversational dataset translated from Chinese Reading Comprehension exam. It includes multi-turn dialogues. It includes annotations for the task of response selection in multi-turn dialogues.

**CIDER** (Ghosal et al., 2021) is a conversational dataset annotated for the task of dialog explanation. Additionally it includes annotation for tasks of span extraction, common sense inference and multiple choice span selection.

**DREAM** (Sun et al., 2019) is a multiple choice question answering dataset. Documents are conversations selected from the English-as-a-foreign-language examinations and questions are created by human experts.

**ReDocRED** (Tan et al., 2022) is a revision of the DocRED dataset which is widely used for document level relation extraction. It includes annotation for 95 different relations.

**DDRel** (Jia et al., 2020) is a dataset for interpersonal relation classification in dyadic conversations. The conversations are extracted from movie scripts in IMSDB and they include annotation for 13 classes of relations.

**DialogRE** (Yu et al., 2020) is a human annotated conversations extracted from Friends sitcom. It includes annotation for 36 relation types between a pair of arguments in a dialogue.

## A.2 ADAPTING TASK-CENTRIC TRANSFER MEASURES

As mentioned in 3, T-MEASURE is the first data-centric transfer measure in zero-shot setting. In this section, we create semantically equivalent data-centric measures based on previous task-centric measures. Particularly, we create data-centric Pairwise Annotation Representation Comparison (PARC) based on the task-centric PARC (Bolya et al., 2021).

**Pairwise Annotation Representation Comparison (PARC)** is a measure to compute the transferability of the model during task transfer i.e. for a trained model $\phi$ with parameters $(T_{src}, D, \beta)$, PARC estimates the transferabilty of $\phi$ on a target task $T_{trg}$. To compute the transferabiltiy it requires a labeled set of the dataset $D$ for $T_{trg}$ which they refer to as the target probe set $\mathcal{P}$. We represent probe set $\mathcal{P}$ as a set of pairs $(x, y)$ where $x$ is a datapoint in $D$ and $y$ is its label for $T_{trg}$. PARC computes the following distance matrices given the probe set $\mathcal{P}$:

$$A_\phi = 1 - corrcoef(f(x)) \ , \ A_{trg} = 1 - corrcoef(g(y)) \tag{14}$$

where correcoef is the Pearson corelation matrix, $f(x)$ is the feature vector of the datapoint $x$ and $g(y)$ is a function of the label of dataset. e.g. for the task of classification $g$ can be the one-hot vector of the classes. Then they compute the transfer estimation as:

$$PARC(\phi, \mathcal{P}) = Spearmanr(\{A_\phi[i,j] : i < j\}, \{A_{trg}[i,j] : i < j\}) \tag{15}$$

PARC assumes that model transferability is higher if datapoints with similar features have similar label for the target task. Therefore it estimates the corelation between distance matrices of datapoint in the feature space and in the target label space.

We use the similar idea of pairwise distances and create data-centric PARC in zero-shot setting. For a trained model $\phi$ with parameters $(T, D_{src}, \beta)$, data-centric PARC, estimates the transferabiltiy of $\phi$ on target dataset $D_{trg}$. Let $\mathcal{P}$ be an unlabeled probe set from $D_{trg}$. We create distance matrices:

$$A_\phi = 1 - corrcoef(R(x)) \ , \ A_{trg} = 1 - corrcoef(\phi(x)) \tag{16}$$

Where $x$ is a datapoint in the probe set and $\phi(x)$ is the models output for the datapoint $x$ and $R(x)$ is the representation of the datapoint in a dataset representation space $R$. Similar to the task-centric PARC, we compute the transferability as follows:

$$PARC_{data}(\phi, \mathcal{P}) = Spearmanr(\{A_\phi[i,j] : i < j\}, \{A_{trg}[i,j] : i < j\}) \tag{17}$$

We implement different versions of PARC by assigning different representation space $R$ and probe sets.

## A.3 Model Selection Experiment

In this section, we include the additional tables used to create the final model selection evaluation table. Tables 4, 5, 9, 11 contain the ground truth of models performance in zero-shot transfer for different tasks. Table 6, 7, 8, 10 contains the T-Measure values assiged for each source task give a target dataset.

Table 4: Accuracy of the transferred models for ERC.

| Target Dataset | Source Dataset | | |
|:---:|:---:|:---:|:---:|
| | Empathetic | DailyDialog | Friends |
| Empathetic | 0.44 | 0.31 | 0.28 |
| DailyDialog | 0.11 | 0.8 | 0.35 |
| Friends | 0.18 | 0.49 | 0.38 |

Table 5: Accuracy of the transferred models for QA.

| Target Dataset | Source Dataset | | |
|:---:|:---:|:---:|:---:|
| | DREAM | CIDER | Friends |
| DREAM | 0.43 | 0.54 | 0.58 |
| CIDER | 0.44 | 0.99 | 0.97 |
| Friends | 0.48 | 0.83 | 0.88 |

## A.4 Results

Table 6: T-Measure for ERC.

| Target Dataset | Source Dataset | | |
|---|---|---|---|
| | Empathetic | DailyDialog | Friends |
| Empathetic | 0.50 | 0.44 | 0.19 |
| DailyDialog | 0.26 | 0.44 | 0.18 |
| Friends | 0.26 | 0.47 | 0.24 |

Table 7: T-Measure for QA.

| Target Dataset | Source Dataset | | |
|---|---|---|---|
| | DREAM | CIDER | Friends |
| DREAM | 1.13 | 0.56 | 0.63 |
| CIDER | 0.62 | 0.57 | 0.71 |
| Friends | 0.61 | 0.65 | 0.59 |

Table 8: T-Measure for RC.

| Target Dataset | Source Dataset | | |
|---|---|---|---|
| | DDRel | DialogRE | ReDocRED |
| DDRel | 0.27 | 1.80 | 0.43 |
| DialogRE | 0.08 | 1.76 | 0.35 |
| ReDocRED | 0.13 | 1.83 | 0.51 |

Table 9: Accuracy of the transferred models for RC.

| Target Dataset | Source Dataset | | |
|---|---|---|---|
| | DDRel | DialogRE | ReDocRED |
| DDRel | 0.29 | 0.49 | 0.23 |
| DialogRE | 0.07 | 0.76 | 0.8 |
| ReDocRED | 0.01 | 0.98 | 0.99 |

Table 10: Accuracy of the transferred models for RS.

| Target Dataset | Source Dataset | | | | |
|---|---|---|---|---|---|
| | DailyDialog | Friends | Empathetic | PersonaChat | Casino |
| DailyDialog | 0.72 | 0.76 | 0.75 | 0.72 | 0.61 |
| Friends | 0.44 | 0.75 | 0.60 | 0.72 | 0.62 |
| Empathetic | 0.40 | 0.63 | 0.63 | 0.69 | 0.70 |
| PersonaChat | 0.56 | 0.73 | 0.61 | 0.81 | 0.59 |
| Casino | 0.40 | 0.64 | 0.69 | 0.76 | 0.74 |
| DailyDialog++ | 0.87 | 0.73 | 0.71 | 0.70 | 0.70 |
| MuTual | 0.63 | 0.66 | 0.59 | 0.72 | 0.73 |

Table 11: Accuracy of the transferred models for RS.

| Target Dataset | Source Dataset | | | | |
|---|---|---|---|---|---|
| | DailyDialog | Friends | Empathetic | PersonaChat | Casino |
| DailyDialog | 1 | 0.99 | 0.99 | 0.99 | 0.99 |
| Friends | 0.99 | 0.98 | 0.9 | 0.96 | 0.95 |
| Empathetic | 1 | 1 | 1 | 1 | 1 |
| PersonaChat | 1 | 0.99 | 0.99 | 1 | 1 |
| Casino | 0.94 | 0.89 | 0.89 | 0.9 | 0.97 |
| DailyDialog++ | 1 | 1 | 1 | 1 | 1 |
| MuTual | 0.74 | 0.72 | 0.71 | 0.71 | 0.72 |

Table 12: The result of model selection based on different transfer measures. The reported numbers are the Kendall-$\tau$ of the ranked models based on the given transfer measure. The size of the probe set in this experiment for all targets is 100. The PARC(SBERT) is the PARC based model that uses SentenceBERT for data representation. PARC($R_{src}$) is a PARC based model that uses a source dataset specific representation space. PARC(TRG) is the version which considers the point-wise co-relation between target and source in computing the transfer measure.

| Task | Target Dataset | Naive | V-Usability | T-Measure | PARC(SBERT) | PARC($R_{src}$) | PARC(TRG) | TM |
|---|---|---|---|---|---|---|---|---|
| **Response Selection** 6 Models | DailyDialog | 0.4 | -0.8 | 0.0 | -0.2 | 0.4 | -0.8 | |
| | Friends | -0.2 | 0.6 | 0.0 | 0.0 | -0.4 | -0.2 | |
| | Empathetic | 0.0 | -0.4 | 0.8 | -0.6 | 0.4 | 0.4 | |
| | PersonaChat | 0.2 | -0.6 | 0.0 | 0.6 | -0.2 | 0.4 | |
| | Casino | 0.6 | -0.6 | 0.2 | 0.4 | 0.6 | 0.4 | |
| | DailyDialog++ | 0.0 | 0.4 | 0.6 | -0.2 | -0.4 | -0.2 | |
| | MuTual | 0.0 | 0.4 | -0.6 | -0.2 | -0.4 | 0.4 | |
| | AVG $\tau$ | **0.14** | -0.14 | **0.14** | -0.02 | 0.09 | 0.05 | |
| | Relative F1 | 0.994 | 0.994 | 0.949 | 0.913 | 0.904 | 0.904 | |
| **Emotion Recognition** 3 Models | Empathetic | 0.33 | -0.33 | -0.33 | -1.0 | -1.0 | -1.0 | |
| | DailyDialog | -0.33 | 0.33 | 0.33 | 1.0 | 1.0 | 1.0 | |
| | Friends | 0.33 | -0.33 | 1.0 | -1.0 | -1.0 | 1.0 | |
| | AVG $\tau$ | 0.11 | -0.11 | **0.33** | -0.33 | -0.33 | **0.33** | |
| | Relative F1 | 0.901 | 0.501 | 0.637 | 0.501 | 0.37 | 0.803 | |
| **Question Answering** 3 Models | Dream | 0.33 | 1.0 | 1.0 | 0.33 | 0.33 | 0.33 | |
| | CIDER | 1.0 | 0.33 | 0.33 | 0.33 | 0.33 | 1.0 | |
| | Friends | 0.33 | 1.0 | 1.0 | 0.33 | -0.33 | -0.33 | |
| | AVG $\tau$ | 0.55 | **0.77** | **0.77** | 0.33 | 0.11 | 0.33 | |
| | Relative F1 | 0.960 | 0.962 | 0.966 | 0.943 | 0.924 | 0.966 | |
| **Relation Classification** 3 Models | DDRel | 0.66 | 1.0 | 1.0 | 0.66 | 0.66 | 0.33 | |
| | DialogRE | 0.0 | 0.33 | 0.33 | 1.0 | 0.66 | 1.0 | |
| | ReDocRED | 0.0 | 0.33 | 0.33 | 0.66 | 0.66 | 0.33 | |
| | AVG $\tau$ | 0.22 | 0.55 | 0.55 | **0.77** | 0.66 | 0.55 | |
| | Relative F1 | 0.823 | 0.979 | 0.979 | 0.556 | 0.843 | 0.559 | |

