# OpenReview forum: "T-Measure: A Measure for Model Transferabilty"
_ICLR.cc/2024/Conference — Submitted to ICLR 2024_

### Official Review · Reviewer_fP8i · 2023-10-29

**Soundness:** 2 fair
**Presentation:** 2 fair
**Contribution:** 2 fair
**Rating:** 3
**Confidence:** 4

**Summary:**

The paper addresses a pertinent challenge in the AI modeling domain, which is model selection for transfer learning. With numerous models available in repositories, each trained on different source data, determining the most appropriate model for a particular task becomes a significant challenge. The authors frame this selection issue as "label-agnostic model selection" - the idea of choosing an efficient model for a specific domain without access to labeled target data.

The main contribution is the introduction of the "T-Measure" as a quantitative tool to estimate the transferability of models based on their training data from the source domain. This measure assesses the transferability by evaluating distributional characteristics of the source domain's data, target domain's data, and the fundamental performance of a task-specific model. The measure aims to provide a ranking of models based on their potential performance on unlabeled target domains.

The authors also touch upon modifying existing task-centric measures to focus on the data, contrasting these against their T-Measure. Through experimental validation on 4 tasks and 11 datasets, they conclude that the T-Measure outperforms baselines in model ranking.

**Strengths:**

1. The idea is clearly stated in the paper, and it is easy to follow. The main idea is to compare the similarity of source data and target data.

2. The problem this paper explores is important, especially in these days where new models come out everyday.

3. The experiments focus on language data, a less explored data type in previous literature.

**Weaknesses:**

1. **Over-simplification of Transferability Factors**: The paper's approach to the complex problem of model transferability is overly reductive. As highlighted in Figure 2, there exists a multitude of factors that can significantly influence transfer performance. The paper seems to narrow its focus primarily on data, while ignoring other crucial aspects. The rationale behind this selective approach is not sufficiently justified. A comprehensive model transferability analysis should account for the interplay of multiple factors, and not just a singular dimension.

2. **Unrealistic Experimental Setting**: The experimental setup employed in this study raises concerns about its applicability in real-world scenarios. Specifically, the paper assumes that models are trained, transferred, and assessed all within the confines of a single task. This is in stark contrast to common practice, where a pre-trained model, developed for one task, is often adapted to suit an entirely different task. Such a limitation restricts the practical utility of the proposed T-Measure.

3. **Opaque Methodological Details**: The paper's exposition on its methodology is not clear in details. For instance, while it uses self-supervised learning for data representation, there's a conspicuous absence of details regarding its implementation. Questions arise about how hyper-parameters for self-supervised learning were chosen and their potential impact on the study's outcomes. The sensitivity of the proposed measure to different self-supervised learning algorithms and training settings remains unclear.

4. **Lack of Rigor in Presentation and Design Choices**: The manuscript frequently resorts to phrases like "intuitively," which undermine the scientific rigor expected of such a study. Additionally, many design choices appear arbitrary, with little to no justification provided. This raises concerns about the robustness and generalizability of the study's conclusions. Specifically, it remains uncertain whether the paper's findings would remain consistent across changes in model architectures, tasks, transfer methods, or self-supervised learning techniques.

In conclusion, while the paper introduces a new T-Measure, its presentation and methodological approach have much room for improvement. A rigorous, comprehensive, and transparent exploration of model transferability is crucial for its findings to hold relevance and value in the broader AI research community.

**Questions:**

1. Can authors summarize and refine the contributions part? The "contribution" of the introduction seems to only summarize each section, not really "contribution".

2. It is not clear if this paper targets at domain adaptation or model fine-tuning. The paper mentions that it focuses on the case where source and target shares the same task, which is domain adaptation actually. If the paper further focuses on the case where Dtrg is not accessible during model selection, it is unsupervised domain adaptation I think. And if the paper deals with selecting source dataset, it is multi-source unsupervised domain adaptation. The literature papers it mentions like Bao et al., 2019; Nguyen et al., 2020; Tran et al., 2019 are all for fine-tuning. They have access to target labels, and they tried different ways to use the target labels. The authors should have a thorough literature review to put this paper into an appropriate position in the literature.

3. There are several incomplete sentences, which I would like the authors to clarify:

- "While some recent body of work sugguest that source dataset is an important factor in transfer: (Zhao et al., 2022) suggest that some datasets are intrinsically harder than others and (Ethayarajh et al., 2021) show that training datasets have different amount of useful information for trained models." in the introduction.
- "when the task, model architecture and transfer method are invariant in the transfer setting" after Figure 2.
- "6 presents boxplots of ranking performance" in Page 8. It should be "Figure 6" I think.

**Details Of Ethics Concerns:**

No.

---

> ### Author Response · Authors · 2023-11-23
>
> We appreciate your constructive feedback. Here is our response to the reviewers questions:
> 1. Our contributions are as follows: We introduce the concept of label-agnostic model selection, presenting a method to estimate the transfer measure and employing it for label-agnostic model selection. The introduced method for estimating transferability is unsupervised and relies on representation learning, making it generalizable and applicable across various domains. Notably, our work is the first to estimate zero-shot transferability, and we adapt measures previously used for transferability estimation in a few-shot setting.
> We compare the adapted measure with our proposed measure (T-measure) and demonstrate the superiority of our introduced transferability estimation through experiments conducted on 11 datasets across 4 tasks.
> 2. The broader problem of model selection can be distilled into either domain adaptation or model fine-tuning based on variables. Consequently, we have incorporated these perspectives into our related works. As correctly pointed out by the reviewer, in cases where the sole variant is the dataset (source and target datasets), our approach aligns with domain adaptation.
> 3. We appreciate the suggested references and related work topics. In response, we have included domain adaptation in the revised version of the paper to provide a comprehensive perspective in addressing your concerns.

---

### Official Review · Reviewer_TVL6 · 2023-10-30

**Soundness:** 3 good
**Presentation:** 2 fair
**Contribution:** 2 fair
**Rating:** 3
**Confidence:** 3

**Summary:**

A new transfer metric is proposed to evaluate target task in zero-shot accuracy setting, when no target labels are available. The new measure estimates the closeness of the target dataset to various source datasets and chooses the model trained on the closest one. The pointwise mutual information metric is used to compute the closeness of the datasets, after learning a suitable representation space on the labeled source dataset. Results have been computed on various dataset settings for the tasks ranging from response selection, emotion recognition, question answering and relation classification.

**Strengths:**

- The paper addresses a very pertinent and important problem of model selection when no target labels are available during model selection (zero-shot model selection).

- The explanation for various types of transfer is presented well in Sec 2.

**Weaknesses:**

- This paper makes a key assumption that the source datasets is decomposable, which might not always be the case. Therefore, the generality of the approach to other datasets is not guaranteed, limiting its impact. Also, they assume that the source dataset is always available during model selection, which is generally not true if we consider the recent trend of foundation models where the source datasets are not available but only source models are released. Lastly, although a minor assumption, it might not always hold that the source and target tasks are same. For example, MLM (masked-langauge-modeling) has been shown to be universally applicable to many downstream tasks for few shot transfer.

- The paper, at its core, essentially evaluates which of the source dataset is the closest to a particular target dataset. In this sense, the authors must also include comparisons with several works in domain adaptation (DA) and robust optimization literature. For example, several measures like A-distance and H-divergence [1] also need to be included. In general, several works in UDA litearture need to be cited and discussed.

- It is really not clearly explained why triplet loss is chosen to learn a suitable representation space. Does the representation learnt using source labels work equally well? Can we also use contrastive loss with multiple negatives? This aspect should be studied in much greater detail.

- The paper states that they use Sentence-BERT for the initial representation. Does this effect the evaluation in any way? For example, does the dataset used to train Sentence-BERT change the ease of transfer to related target domains?

- I am curious to know why the authors chose to provide examples related to vision datasets and vision tasks (in Fig 2), while all the evaluation is done using NLP datastes. Even more so, Fig 1-4 are all really not explaining anything related to the problem or method, and could be improved to illustrate your idea better.

#### Minor

- Sec 3.2, Step1: Did you mean argmin in the equation?

- For completeness, PVI should be explained using an equation or such, since not all readers might be familiar with the term (I am not!).


[1] Ben-David, Shai, et al. "A theory of learning from different domains." Machine learning 79 (2010): 151-175.

**Questions:**

I feel the intuitions behind several choices in the paper could be presented better and the evaluation could be made more extensive by comparing with other measures of distance, representation learning methods and transfer settings. I request the authors to address my concerns above and I'd be happy to raise the rating.

---

> ### Author Response · Authors · 2023-11-23
>
> We thank the reviewer for their constructive feedback.
> Our proposed method for learning a representation space for the dataset relies on information about the dependency among datapoints in the dataset. While the dataset doesn't necessarily have to be decomposable, our model remains applicable as long as we can identify dependencies among data points.
> Acknowledging that source data might not always be available during the model selection phase, we aim to enhance the broad applicability of the T-measure. We plan to learn a model representation, estimating the transfer measure using the model instead of relying on the availability of the source dataset or other confounding transfer factors. In response to the observation that tasks in the source and target may not always be similar, our current focus in this paper is on models and zero-shot transfer, where the transfer function is an identity function. This function requires the task space to be similar for the source and target. We intend to extend our work to cover other functions in future research.
> We appreciate the references you provided, and we have duly included them in the related work section.
> The choice of triplet loss is grounded in its ability to serve as a proxy for modeling the dependency of datapoints in the dataset without supervision. In triplet loss, anchor and positive points are considered dependent, while anchor and negatives are independent. Unlike learning representations using labels, which limits the representation to specific tasks, our approach learns a dataset representation that can be applied to multiple tasks (e.g., using daily dialog for emotion recognition and response selection). In our experiments, we employed multiple negative datapoints for contrastive loss (19 points for each anchor and positive).
> It's important to note that the introduced T-measure is not confined to NLP; it is applicable to vision datasets as well, therefore, the examples in Figure 2 are for the vision domain. Although we plan to include vision datasets in our future work, the limited number of publicly accessible vision datasets has been a constraint. Figure 1 provides an illustration of the general model selection problem, and we have updated Figures 3 and 4 to better convey the ideas behind our method.
> We addressed the minor issues in the new revision of the draft.

---

### Official Review · Reviewer_7czX · 2023-11-01

**Soundness:** 3 good
**Presentation:** 2 fair
**Contribution:** 2 fair
**Rating:** 3
**Confidence:** 3

**Summary:**

The paper introduces a metric for measuring transferability score across datasets, called T-measure. The measure introduces a two step process to measure the transferability score - 1. selecting a subset similar to target dataset, 2. using PVI of datapoints and aggregating to measure the final score. Experiments are shown on different datasets, comparing with existing techniques, which demonstrate that the method performs better on average than competing scoring methods.

**Strengths:**

The model has been evaluated on different datasets, showing comparable or better performance than competing methods. The domain of testing includes classification, emotion recognition, question/answering, etc. Previous methods were focused on classification methods only.

**Weaknesses:**

The current draft of the paper has major weaknesses as listed below:

1. Equations are not numbered in the paper, which is lowering the readability of the paper. For example, equations in Section 2.1 and 2.2 are outputting \phi^* model, on the first read its not clear what is the difference between the two equations. Similarly, in Section 3.2, PVI computation of data samples is not provided, hence the motivation for Step 2 is not clear from the draft. In the results section, Table 4 and 6 are referenced for results, but these tables are present in Appendix, not the main section of the paper. Not sure if it violates the page length limit in the conference.

2. The proposed method is an increment over the (Ethayarajh et al., 2021) which introduced V-Usabiltiy and Pointwise V-Information(PVI). The current paper aggregates the PVI over a subset of samples from source, which are similar to target. Are there other functions which can be considered for aggregation, can authors provide justifcation for selecting this particular method. Also, it is not clear how this applies for a generic transferred model (α(T, Dtrain_trg , ϕi)). The paper mentions that α is identity in Section 2.4, but its not clear otherwise.

3. In Table 3, it is not clear what average Kendell-tau distance signifies? There is no point averaging across datasets. Can authors specify if the model is outperforming other methods on individual datasets. Also, references for competing methods are not provided in the table.

**Questions:**

Weakness section has questions about the paper, which should be answered in the rebuttal.

---

> ### Author Response · Authors · 2023-11-23
>
> We greatly appreciate your constructive comments. In the revised version, we will organize the equations and assign numbers accordingly.
>
> Regarding the difference between equations 2.1 and 2.2: Equation 2.1 represents the general equation for model selection. However, in equation 2.2, the model is selected among transferred models under a specific transfer method (\alpha). The assumption is that the set of models {\phi} is transferred to a set of {\phi’} models under the transfer method \alpha. The objective of model selection becomes choosing the best transferred model {\phi’}. To enhance clarity, we have added the definition of PVI.
>
> In reference to the Kendall-Tau ranking measure in Table 3, it illustrates how the ranking of models based on the selected transfer measure aligns with the actual performance of the models on the target. A higher Kendall-Tau value indicates a more accurate ranking of models, reflecting a ranking more similar to the models' actual performance on the target. We have updated the table caption to provide clarity on this aspect.
>
> Furthermore, we have revised the table to emphasize the best transfer measure for each target. Notably, for six targets, the T-measure yields a higher ranking compared to other metrics (naive: 4, V-usability: 5, PARC: 4). In the initial paragraph of Section 4.3, where baselines are described, we have included the relevant references. It is crucial to note that, as the first zero-shot data-centric transfer measure, the baselines were modified to align with the T-Measure.
>
> Thank you for your attention to detail and suggestions.

---

### Official Review · Reviewer_Umyu · 2023-11-01

**Soundness:** 3 good
**Presentation:** 3 good
**Contribution:** 2 fair
**Rating:** 5
**Confidence:** 3

**Summary:**

This paper introduces a new metric for estimating model transferability in zero-shot transfer learning. The authors cast the problem as label-agnostic model selection, which aims to select the best model on a target dataset without annotations. The authors propose to learn a representation space aligned with each source dataset using contrastive learning on triplets which captures the dependency between data. Experiments on multiple tasks and datasets demonstrate the effectiveness of the proposed metric for model selection compared to other baselines.

**Strengths:**

1. The paper studies an important practical problem of model selection for transfer learning in the absence of labeled data.

2. The proposed data-centric transferability measure based on source/target dataset similarity is interesting.

3. T-measure shows consistent benefit over baselines on diverse tasks and datasets.

**Weaknesses:**

1. The proposed metric requires access to the source data. However, the source data may not always be available during fine-tuning.
2. The number of pre-trained models is quite limited. According to Table 3, the metric is only evaluated on 3 or 6 models. It remains unclear whether the metric can be extended to a large number of pre-trained models.

**Questions:**

None

---

> ### Author Response · Authors · 2023-11-22
>
> We appreciate your feedback. We agree that the source data might not always be available during the model selection phase. To overcome this issue and make the T-measure broadly applicable, we plan to learn a model representation, i.e., instead of estimating the transfer measure using the source dataset, estimate it using the model, which does not require the availability of the source dataset and other confounders of transfer.
> The reason behind the limited number of source models is that we fixed all of the parameters and factors(model architecture, transfer method, ...) in model selection except the source dataset so that the results can be fairly compared together. In this paper, our focus is to
>  solely estimate the effect of the source dataset on the transfer. And the number of models in such a setting becomes limited to the number of available datasets for that task under the same conditions. For example, we found only 3 annotated datasets for utterance-level conversation emotion recognition with 6 basic emotions.

---

### Meta-Review · Area_Chair_67X1 · 2023-12-02

**Metareview:**

This paper considers the “zero-shot” transfer learning scenario where the transfer method uses the same model before transfer. The target training dataset is unlabeled. The paper introduces the model selection problem in this zero-shot transfer learning scenario and proposes the “T-measure” that quantifies the model transferability. Experiments are conducted on multiple datasets and several baseline scoring methods to evaluate the effectiveness of the proposed T-measure in model ranking.

**Justification For Why Not Higher Score:**

Restrictive assumptions:
- The proposed metric requires source data, which is not obtainable sometimes.
- The method requires a decomposability assumption and requires capturing the dependency structure of data points, which is not an easy task.
- The source tasks and the downstream tasks have to be the same.
- Lack of transfer measures for generic transferring techniques beyond zero-shot.

Limited experimental validations, unconvincing results, and lack of evaluated datasets.
Paper writing issues, such as lack of rigor and lack of methodological details.

**Justification For Why Not Lower Score:**

N/A

---

### Decision · Program_Chairs · 2024-01-16

Reject